

# Association of triglyceride-glucose index with postoperative recovery-related factors in type II cesarean scar pregnancy treated by transvaginal surgery

Xiaoyan Li, Shuhua Liu and Dehong Liu

Anhui Provincial Women and Children's Medical Center, Hefei Maternal and Child Health Hospital, Obstetrics and Gynecology Department, Hefei, Anhui, China

## ABSTRACT

**Background:** Type II cesarean scar pregnancy (CSP) is a rare and serious condition. The triglyceride-glucose index (TyG) may influence postoperative recovery-related factors (PRRF) as a marker of insulin resistance. This study investigates the relationship between TyG and PRRF in patients with type II CSP undergoing transvaginal surgery.
**Methods:** Forty-five patients treated from January 2019 to September 2023 were analyzed. Data on surgical duration, intraoperative blood loss, length of stay, hospitalization costs, and recovery were collected and assessed against TyG levels.
**Results:** A significant correlation was found between TyG and hospitalization costs ($\beta = -677.5$, $p = 0.02$). Higher TyG levels were associated with shorter hospital stays and lower costs, while no significant associations were observed with other factors.
**Conclusion:** TyG is associated with hospitalization costs and length of stay in type II CSP patients, suggesting its potential to predict healthcare resource utilization. Further research is needed to explore its role in surgical risk assessment.

## INTRODUCTION

Cesarean scar pregnancy (CSP) is a rare but potentially life-threatening form of ectopic pregnancy, particularly type II CSP, characterized by the implantation of the gestational sac either partially or completely within the uterine scar tissue from a previous cesarean section (*Gonzalez & Tulandi, 2017*). As cesarean rates have risen, the incidence of type II CSP has likewise increased, and variations in treatment strategies and perioperative management can profoundly affect reproductive health and subsequent pregnancy outcomes in women of childbearing age (*Calì et al., 2018*; *Morlando, Conte & Schiattarella, 2015*). Accordingly, careful selection of therapeutic approaches and vigilant monitoring before and after intervention are essential for optimizing outcomes in patients with type II CSP.

Type II CSP is primarily managed through surgical intervention, often combined with adjuvant therapies such as medication or uterine artery embolization (UAE). Various surgical approaches are currently employed in clinical practice, including

Corresponding author
Dehong Liu, klorado@sina.com

ultrasound-guided aspiration, hysteroscopic resection, and lesion excision with uterine repair (*Huo et al., 2023*). Transvaginal surgery is one of the most common procedures for the treatment of type II CSP (*Zhang et al., 2015*). A recent cohort study demonstrated that, compared with UAE, transvaginal surgery was associated with shorter hospital stays and more rapid menstrual recovery; however, these favorable outcomes showed no correlation with serum β-hCG levels, gestational age, or maximal gestational sac diameter at diagnosis (*Chen et al., 2015*). Whether postoperative recovery after transvaginal surgery for type II CSP is influenced by other biochemical markers remains to be determined.

The outcomes of surgery and postoperative recovery may be influenced by various factors, including the patient's metabolic status (*Helander et al., 2019*). The triglyceride-glucose index (TyG), a simple measure of insulin resistance, has increasingly gained recognition in predicting metabolic disorders and cardiovascular events (*Alizargar et al., 2020*). Studies have indicated that the TyG index may be closely associated with surgical-related outcomes, including postoperative recovery, complications, and duration of hospital stay (*Sun et al., 2024*). For instance, research from Turkey has suggested a significant correlation between the TyG index and hospital stay in patients undergoing partial hip arthroplasty for femoral neck fractures (*Astan & Balta, 2024*). Therefore, exploring the role of the TyG index in postoperative recovery may provide new insights and evidence to improve clinical management strategies.

Patients with type II CSP typically include women who have undergone cesarean delivery, and during pregnancy, they may exhibit insulin resistance, elevated triglycerides and cholesterol levels, abnormal blood glucose, and increased inflammatory markers (*Vejrazkova et al., 2014*; *Lain & Catalano, 2007*; *Nadeau-Vallée et al., 2016*). The extent of these metabolic derangements may correlate with the severity of postoperative recovery and complication risk. Accordingly, we hypothesize that the TyG index, as a practical surrogate for insulin resistance, may to some extent capture these metabolic disturbances and serve as a predictor of surgical outcomes and complications in patients with type II CSP treated *via* transvaginal surgery.

In this study, the surgery duration, intraoperative blood loss, length of hospital stay, hospitalization costs, postoperative vaginal bleeding duration, and time to menstrual recovery were defined as postoperative recovery-related factors (PRRF). This study aims to investigate the relationship between the TyG index and PRRF in patients with type II CSP who undergo transvaginal surgery, evaluating its potential associations with surgical duration, intraoperative blood loss, length of hospital stay, hospitalization costs, duration of postoperative vaginal bleeding, and time to menstrual recovery.

## METHODS

### Study population

Inclusion criteria were as follows: (1) Preoperative ultrasonographic findings consistent with the diagnostic criteria for type II cesarean scar pregnancy (CSP) as outlined in the Expert Consensus on Diagnosis and Treatment of Cesarean Scar Pregnancy (2016), a guideline established in China (*Huo et al., 2023*). These criteria include gestational sac implantation partially at the cesarean scar, with partial or most of the sac located within

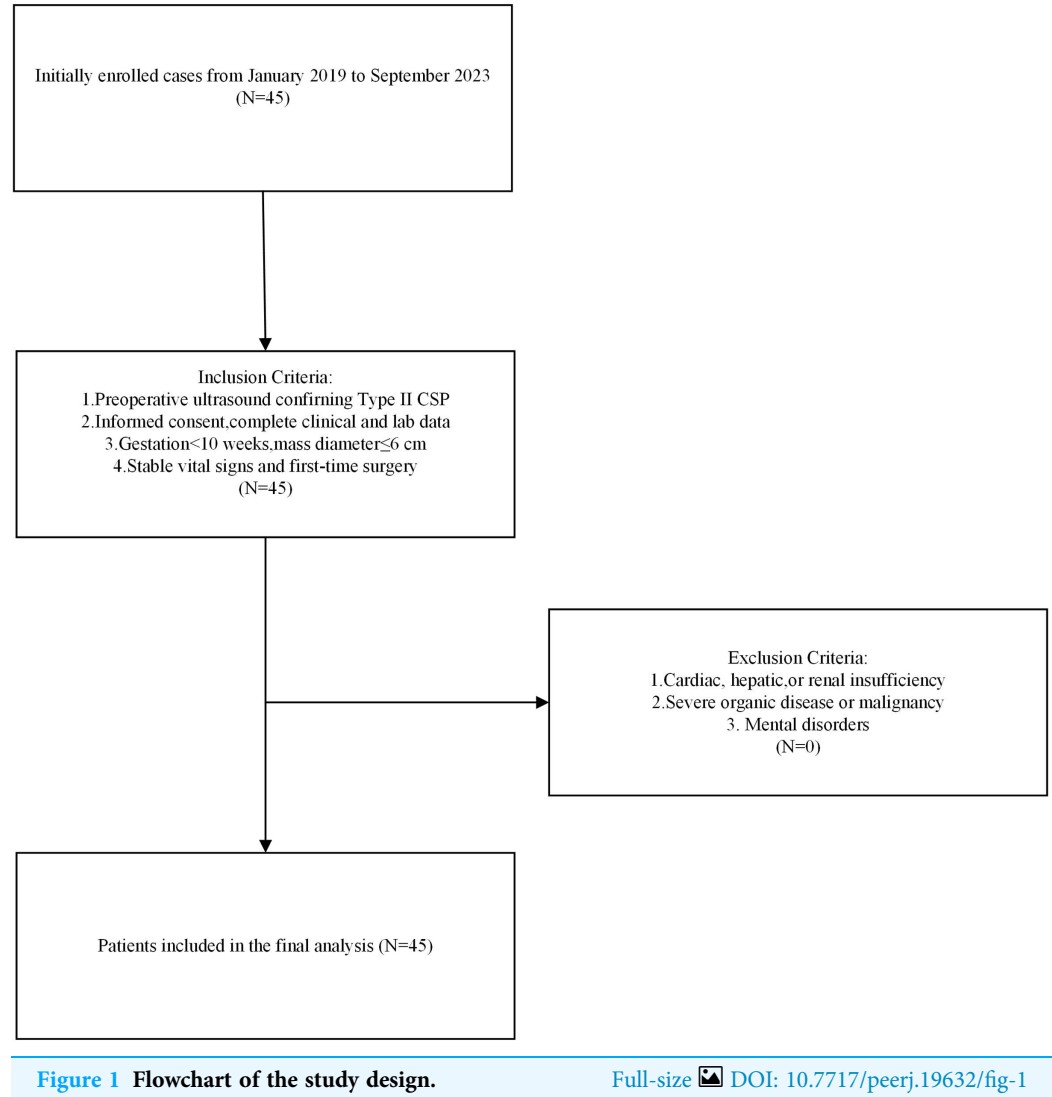

**Figure 1 Flowchart of the study design.**

the uterine cavity, and in some cases extending to the fundus. The gestational sac is notably deformed, elongated, and sharply angled at the lower end. The myometrium between the gestational sac and the bladder is thinned to ≤3 mm. Color Doppler flow imaging (CDFI) reveals trophoblastic blood flow signals at the scar site, characterized by low-resistance blood flow; (2) informed consent obtained before surgery, with complete clinical and laboratory data; (3) gestation period of fewer than 10 weeks and mean diameter of the mass ≤6 cm; (4) stable vital signs and first-time surgical treatment. Exclusion criteria included: (1) cardiac, hepatic, or renal insufficiency; (2) severe organic disease or malignancies; (3) mental disorders. The flowchart is presented in Fig. 1.

## Data collection and definition

Physicians collected demographic characteristics, medical history, and laboratory test results from the electronic medical record system. All patients were married women with

**Table 1 Clinical and biological characteristics classified by TyG index.**

| Characteristics | T1 (7.11–7.95) ($n$ = 15) | T2 (7.96–8.24) ($n$ = 15) | T3 (8.24–9.51) ($n$ = 15) | $p$-value |
|---|---|---|---|---|
| Age | 32.1 ± 2.5 | 32.1 ± 2.9 | 34.3 ± 4.2 | 0.124 |
| TyG | 7.6 ± 0.3 | 8.1 ± 0.1 | 8.7 ± 0.4 | <0.001 |
| Total cholesterol (mmol/L) | 3.8 ± 1.1 | 4.1 ± 0.9 | 4.2 ± 0.8 | 0.434 |
| High-density lipoprotein (mmol/L) | 1.2 ± 0.2 | 1.3 ± 0.2 | 1.2 ± 0.2 | 0.498 |
| Days since menstruation | 47.1 ± 9.7 | 47.3 ± 7.8 | 48.1 ± 6.0 | 0.936 |
| Number of pregnancies | 4.0 ± 1.5 | 4.4 ± 1.4 | 4.8 ± 2.7 | 0.542 |
| Time since last cesarean section (years) | 5.5 ± 2.3 | 4.7 ± 3.1 | 5.9 ± 2.6 | 0.497 |
| LMP (years) | 16.7 ± 8.5 | 18.1 ± 6.8 | 15.1 ± 5.5 | 0.496 |
| Preoperative HCG level (mIU/ml) | 42,060.1 ± 46,712.6 | 55,574.7 ± 48,662.9 | 46,028.8 ± 43,142.9 | 0.715 |
| Surgery duration (minutes) | 28.9 ± 6.9 | 37.8 ± 9.8 | 29.2 ± 6.9 | 0.005 |
| Intraoperative blood loss (ml) | 16.0 ± 11.1 | 16.3 ± 5.5 | 14.7 ± 12.0 | 0.889 |
| Length of hospital stay (days) | 7.3 ± 1.6 | 6.5 ± 1.4 | 6.3 ± 2.2 | 0.242 |
| Hospitalization cost (CNY) | 8,954.7 ± 907.9 | 8,170.6 ± 900.9 | 8,133.0 ± 697.6 | 0.016 |
| Postoperative vaginal bleeding duration (days) | 10.1 ± 1.3 | 9.8 ± 1.5 | 9.6 ± 1.1 | 0.626 |
| Time to menstrual recovery (days) | 35.1 ± 4.0 | 36.4 ± 2.8 | 36.0 ± 2.9 | 0.567 |

Notes:
Abbreviations TyG, Triglyceride-Glucose index; HCG, Human Chorionic Gonadotropin; CNY, Chinese Yuan; LMP: days from the 1st day of the last menstrual period.
All data are presented as mean ± standard deviation (SD).

no history of smoking or alcohol consumption. The triglyceride-glucose (TyG) index was calculated using the formula: ln[fasting triglyceride level (mg/dL) × fasting glucose level (mg/dL)/2] (*Liu et al., 2024*).

## Statistical analysis

Participants were divided into three groups according to TyG index tertiles: T1, T2, and T3, with variance analysis performed to compare group differences across multiple indicators. Linear correlation analysis was performed between TyG index and PRRF. After adjusting for age, total cholesterol, and high-density lipoprotein, multivariate regression analysis was conducted to assess the relationship between TyG index and PRRF. All statistical analyses were performed using R version 4.4.1, employing two-tailed tests, with significance established at an alpha level of $p < 0.05$.

## Ethical approval and informed consent

The study received ethical approval from the Ethics Committee of Anhui Provincial Women's and Children's Medical Center (approval number: 83220468). Before enrollment, written informed consent was obtained from all participants.

## RESULTS

Table 1 presents the clinical and biological characteristics of participants stratified by the TyG. There were no significant differences in age ($p$ = 0.124), total cholesterol ($p$ = 0.434), high-density lipoprotein ($p$ = 0.498), days from the 1st day of the last menstrual period (LMP) ($p$ = 0.936), number of pregnancies ($p$ = 0.542), or time since the last cesarean section ($p$ = 0.497). However, TyG levels significantly increased across the groups
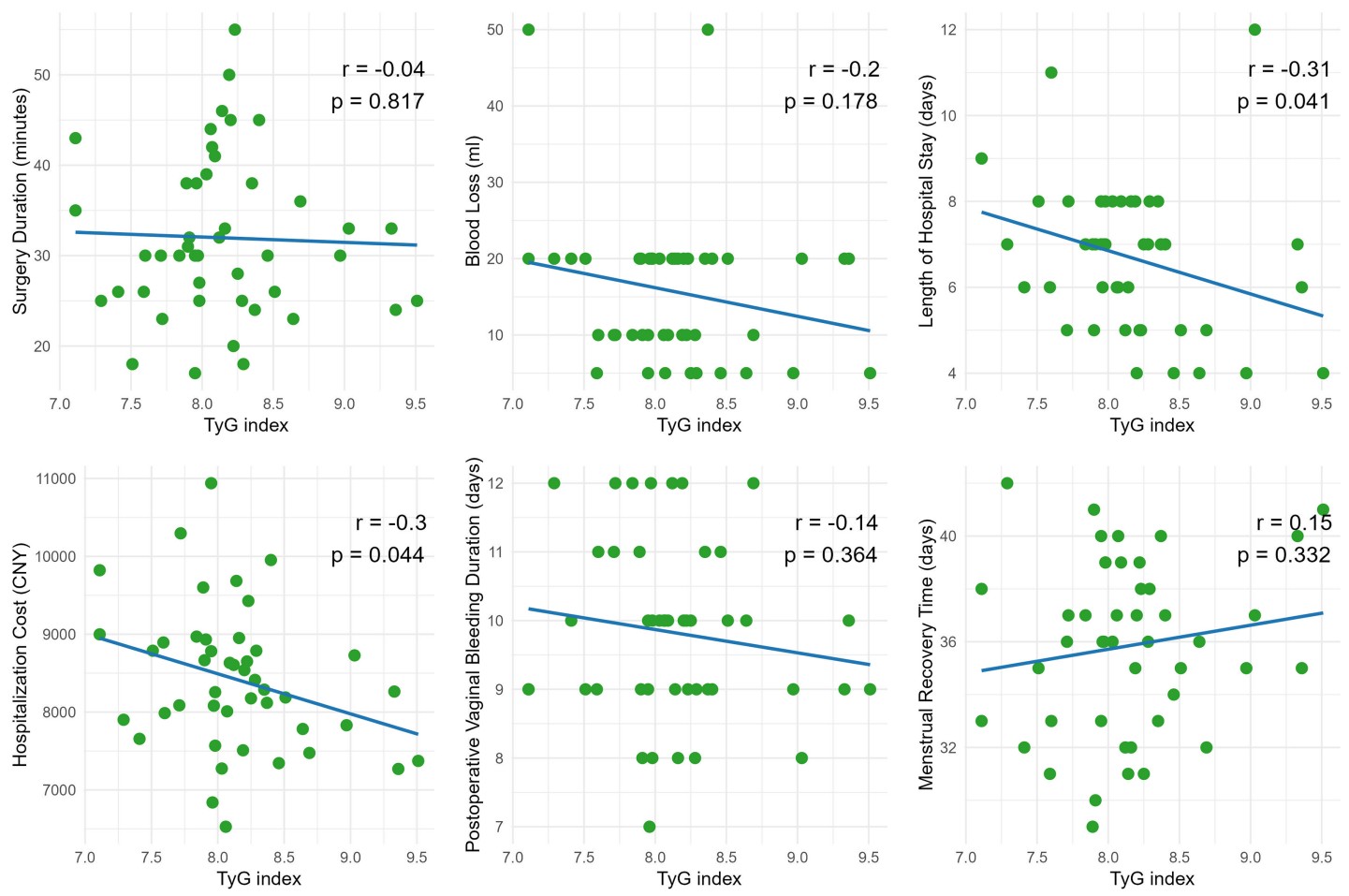

**Figure 2** Linear correlation analysis between the triglyceride-glucose index (TyG) and surgical as well as postoperative recovery indicators.

($p < 0.001$). Surgery duration was considerably longer in T2 than in T1 ($p = 0.005$), and hospitalization costs were substantially lower in T2 than in T1 ($p = 0.016$). Intraoperative blood loss ($p = 0.889$), length of hospital stay ($p = 0.242$), postoperative vaginal bleeding duration ($p = 0.626$), and time to menstrual recovery ($p = 0.567$) did not differ significantly across groups.

Figure 2 presents scatter plots depicting the correlations between the Triglyceride-Glucose (TyG) index and PRRF. No significant correlations were observed between TyG index and surgery duration ($r = -0.04$, $p = 0.817$), intraoperative blood loss ($r = -0.2$, $p = 0.178$), postoperative vaginal bleeding duration ($r = -0.14$, $p = 0.364$), or menstrual recovery time ($r = 0.15$, $p = 0.332$). However, the TyG index showed a statistically significant negative correlation with both length of hospital stay ($r = -0.31$, $p = 0.041$) and hospitalization cost ($r = -0.3$, $p = 0.044$), indicating that higher TyG values were associated with shorter hospital stays and lower hospitalization costs.

The multivariable regression analysis presented in Table 2 explores the association between the TyG index and PRRF, adjusting for age, total cholesterol, and high-density

**Table 2 Multivariable regression analysis of TyG in relation to surgical and postoperative recovery outcomes.**

| Evaluation index | Factor | β Value | Standard error | t-Value | p-Value |
|---|---|---|---|---|---|
| Surgery duration | TyG | −0.09 | 2.94 | −0.03 | 0.98 |
| Surgery duration | Age | −0.24 | 0.47 | −0.52 | 0.61 |
| Surgery duration | Total cholesterol | −0.18 | 2.04 | −0.09 | 0.93 |
| Surgery duration | High-density lipoprotein | 13.55 | 8.49 | 1.6 | 0.12 |
| Intraoperative blood length of hospital stays | TyG | −1.69 | 3.22 | −0.53 | 0.6 |
| Intraoperative blood length of hospital stays | Age | −0.7 | 0.51 | −1.37 | 0.18 |
| Intraoperative blood length of hospital stays | Total cholesterol | −0.92 | 2.23 | −0.41 | 0.68 |
| Intraoperative blood length of hospital stays | High-density lipoprotein | 5.98 | 9.3 | 0.64 | 0.52 |
| Length of hospital stay | TyG | −1.06 | 0.55 | −1.91 | 0.06 |
| Length of hospital stay | Age | 0.13 | 0.09 | 1.46 | 0.15 |
| Length of hospital stay | Total cholesterol | −0.24 | 0.38 | −0.64 | 0.53 |
| Length of hospital stay | High-density lipoprotein | 1.97 | 1.6 | 1.23 | 0.23 |
| Hospitalization cost | TyG | −677.5 | 281.35 | −2.41 | 0.02 |
| Hospitalization cost | Age | 92.8 | 44.95 | 2.06 | 0.05 |
| Hospitalization cost | Total cholesterol | 106.13 | 195.17 | 0.54 | 0.59 |
| Hospitalization cost | High-density lipoprotein | −366.8 | 812.41 | −0.45 | 0.65 |
| Postoperative vaginal bleeding duration | TyG | −0.53 | 0.41 | −1.28 | 0.21 |
| Postoperative vaginal bleeding duration | Age | 0.06 | 0.07 | 0.98 | 0.33 |
| Postoperative vaginal bleeding duration | Total cholesterol | −0.02 | 0.29 | −0.08 | 0.94 |
| Postoperative vaginal bleeding duration | High-density lipoprotein | −0.06 | 1.19 | −0.05 | 0.96 |
| Time to menstrual recovery | TyG | 1.45 | 0.92 | 1.58 | 0.12 |
| Time to menstrual recovery | Age | 0.16 | 0.15 | 1.11 | 0.27 |
| Time to menstrual recovery | Total cholesterol | −1.38 | 0.64 | −2.17 | 0.04 |
| Time to menstrual recovery | High-density lipoprotein | 5.69 | 2.65 | 2.15 | 0.04 |

**Note:**
Abbreviations TyG, Triglyceride-Glucose index.

lipoprotein. TyG was not significantly correlated with surgery duration (β = −0.09, p = 0.98), and no significant associations were observed after adjustment for confounders. Similarly, TyG had no considerable impact on intraoperative blood loss (β = −1.69, p = 0.60), with adjusted factors remaining non-significant. A borderline association was observed between TyG and length of hospital stay (β = −1.06, p = 0.06), although this was not significant after adjustment. Notably, TyG was significantly associated with a reduction in hospitalization costs (β = −677.5, p = 0.02). At the same time, age showed a positive correlation (β = 92.8, p = 0.05), and neither total cholesterol nor HDL had a significant influence. TyG was not significantly associated with postoperative vaginal bleeding duration (β = −0.53, p = 0.21), nor with age, total cholesterol, or HDL in this context. Similarly, TyG did not significantly affect menstrual recovery time (β = 1.45, p = 0.12). These findings indicate a significant association between TyG and reduced hospitalization costs, while other factors were not significantly influenced. Age, total cholesterol, and HDL were primarily associated with hospitalization costs and menstrual recovery time.

## DISCUSSION

This study evaluated the relationship between the TyG and PRRF measures, including surgery duration, intraoperative blood loss, length of hospital stay, hospitalization costs, duration of postoperative vaginal bleeding, and time to menstrual recovery. The study focused on patients with type II CSP undergoing transvaginal surgery. The results indicated a significant correlation between TyG and hospitalization costs among all PRRF studied. At the same time, no clear associations were found with other surgical endpoints, even after adjusting for confounding factors such as age, total cholesterol, and high-density lipoprotein.

The analysis demonstrated no significant correlation between TyG and surgery duration, intraoperative blood loss, postoperative vaginal bleeding duration, or time to menstrual recovery. However, the literature suggests that elevated TyG levels may correlate with adverse clinical outcomes due to their association with insulin resistance and metabolic syndrome (*Tahapary et al., 2022*). Elevated TyG levels may be associated with adverse clinical outcomes in various diseases. For example, some researchers have found that the TyG index can predict the incidence of acute kidney injury in patients with critical heart failure (*Yang et al., 2023*). However, the results of this study do not support a direct correlation between TyG and these perioperative indicators. This result may be attributed to the specificity of cicatricial pregnancy by cesarean section and the relatively consistent surgical techniques used to manage this condition. In addition, the small sample size may also limit the ability to find weak associations between TyG and certain PRRF, such as time to surgery or amount of blood loss.

In the multivariate linear regression analysis, a marginally significant correlation was observed between TyG and length of hospital stay ($p = 0.06$). However, linear correlation analysis revealed a significant negative association between TyG and length of stay ($r = -0.31$, $p = 0.041$). Although this result did not achieve statistical significance, it suggests that higher TyG levels may be associated with a reduction in length of stay, which contrasts with previous studies indicating that insulin resistance and metabolic disorders prolong recovery times in various surgical contexts (*Thorell, Nygren & Ljungqvist, 1999*). This phenomenon may relate to compensatory mechanisms associated with insulin resistance (*Ljungqvist, 2010*). This phenomenon may relate to compensatory mechanisms associated with insulin resistance.

The most notable finding of this study is the significant negative correlation between TyG and hospitalization costs ($\beta = -677.5$, $p = 0.02$). Even after adjusting for age, total cholesterol, and HDL, patients with higher TyG levels exhibited significantly lower hospitalization costs. Notably, age was positively correlated with hospitalization costs ($\beta = 92.8$, $p = 0.05$), while lipid indicators such as total cholesterol and high-density lipoprotein did not significantly influence costs. The reasons for this negative correlation between TyG and hospitalization costs remain unclear. One possible explanation is that patients with higher TyG levels experienced fewer postoperative complications or required fewer postoperative interventions, reducing costs. Another possibility is that metabolic factors associated with higher TyG levels may influence postoperative care and

management, leading to decreased healthcare expenses (*Carli, 2015*). Future research should explore whether TyG-related metabolic factors directly or indirectly affect healthcare costs.

The significant association between TyG and hospitalization costs has important implications for clinical practice and healthcare management. Given that TyG is an easily obtainable marker of metabolic risk, it could predict healthcare resource utilization in complex surgeries such as those for CSP. Some researchers have proposed that TyG may be a valuable risk stratification and management tool in high-risk populations, such as patients with chronic kidney disease and cardiovascular disease (*Ye et al., 2023*). Incorporating metabolic markers like TyG into preoperative risk assessments may assist healthcare providers in better estimating and managing hospitalization costs, thereby improving resource allocation and optimizing patient care.

However, the lack of significant associations between TyG and most PRRF suggests that reliance on TyG alone may be insufficient for predicting perioperative complications in CSP patients. Our results indicate that PRRF such as intraoperative blood loss and postoperative recovery duration appear unrelated to TyG levels, at least within the context of type II CSP. Future studies should better investigate other potential biomarkers or patient characteristics to predict surgical risks and recovery in this population.

Body mass index (BMI), a widely recognized indicator of metabolic health, has been identified as a predictor of insulin resistance (*Cho et al., 2011*), suggesting a potential association between BMI and the triglyceride-glucose (TyG) index. BMI is a prognostic factor for surgical outcomes in bariatric and cardiac surgery (*Gondal et al., 2019*; *Aftab et al., 2014*). In our study, BMI was not included in the analysis due to data availability constraints arising from the retrospective study design. BMI was not consistently recorded in the clinical dataset. Future prospective studies incorporating BMI and other metabolic parameters are warranted to explore their role in predicting obstetric surgical outcomes further. This study has several other limitations that should be acknowledged. First, the relatively small sample size may have limited the ability to detect significant associations between TyG and certain perioperative risk-related factors, such as surgery duration and blood loss. Second, unmeasured confounding factors, such as other metabolic or inflammatory markers, may have influenced the results.

## CONCLUSION

In summary, the results of this study indicate that the triglyceride-glucose index (TyG) is associated with hospitalization costs and length of stay following transvaginal surgery in patients with type II cesarean scar pregnancy. At the same time, no significant correlations were found with other PRRF. These findings suggest that TyG may be a potential biomarker for predicting healthcare resource utilization, although its role in forecasting surgical risks remains unclear. Future research with larger sample sizes is needed to validate these findings and explore the underlying mechanisms of the relationship between TyG and healthcare expenditures.

## ACKNOWLEDGEMENTS

While preparing this manuscript, the authors used ChatGPT to assist in language polishing. After utilizing this tool, the authors thoroughly reviewed and edited the content as necessary and took full responsibility for the integrity and accuracy of the final manuscript.

### Funding

This work was funded by the Anhui Provincial Higher Education Scientific Research Project (Natural Science) (2022AH050674). The funders had no role in study design, data collection and analysis, decision to publish, or preparation of the manuscript.

### Grant Disclosures

The following grant information was disclosed by the authors:
Anhui Provincial Higher Education Scientific Research Project (Natural Science): 2022AH050674.

### Competing Interests

The authors declare that they have no competing interests.

### Author Contributions

- Xiaoyan Li performed the experiments, prepared figures and/or tables, and approved the final draft.
- Shuhua Liu conceived and designed the experiments, analyzed the data, authored or reviewed drafts of the article, and approved the final draft.
- Dehong Liu analyzed the data, authored or reviewed drafts of the article, and approved the final draft.

### Human Ethics

The following information was supplied relating to ethical approvals (*i.e.*, approving body and any reference numbers):

The Ethics Committee of Anhui Provincial Women's and Children's Medical Center provided approval number: 83220468.

### Data Availability

Raw data is available in the Supplemental Files.

### Supplemental Information

Supplemental information for this article can be found online at http://dx.doi.org/10.7717/peerj.19632#supplemental-information.

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
