# Peer review of "Association of triglyceride-glucose index with postoperative recovery-related factors in type II cesarean scar pregnancy treated by transvaginal surgery"

_PeerJ, doi:10.7717/peerj.19632_

## Round 0.1 · original submission · Major Revisions

Although brief in nature, the comments from the reviewers are major in some respects. Reviewer 2 has gone so far as to recommend rejection. Therefore, the authors must address all the concerns (explicit and implied) in detail

Reviewer 1 ·

Basic reporting

The author incorporates data on surgical duration within the manuscript; however, it is inaccurate to categorize factors such as intra-operative blood loss, length of stay, and hospitalization costs as postoperative outcomes.
The Figure 1 was not standardized and needs to be revised.

Experimental design

Presently, there exist two clinical classification methods. The manuscript does not explicitly clarify the classification criteria for Type II CSP.

The primary objective of this study is to investigate the association between the Triglyceride-Glucose (TyG) index and postoperative outcomes in cases of type II cesarean scar pregnancy (CSP). The exclusion of Body Mass Index (BMI) from the study warrants clarification

Validity of the findings

The clinical significance of this study appears to be relatively limited.

Additional comments

Additionally, it is imperative to acknowledge that the treatment strategy for CSP extends beyond transvaginal surgery; therefore, alternative treatment modalities should be addressed in the introduction or discussion sections.

Reviewer 2 ·

Basic reporting

The term "days since menopause" is rather unclear. Perhaps "days from the 1st day of the last menstrual period (LMP)" would be more natural.

Figure 1 could be modified, starting from initially enrolled cases (45 cases) branching to excluded cases (0 cases) via exclusion criteria and finally eligible cases (45 cases).

In Table 1, 'Time since last cesarean section (years)' has been duplicated.

Experimental design

The title of the research is different from the title of the ethical approval document.

Validity of the findings

I cannot understand the raw data results. Why did the "less than 10 ml intraoperative blood loss" cases take 4 to 11 days stay?

And in the case of No. 22, the preoperative HCG value is 4078, although the mean diameter of the lower uterine incision mass is 34mm.

---

## Round 0.2 · Minor Revisions

Thank you for your amendments. You have addressed the comments raised by reviewers in most cases. Some minor amendments are needed for consistency of language, titles, and references. Please see the comments from the reviewers for specific areas to address.

Reviewer 3 ·

Basic reporting

no comment

Experimental design

no comment

Validity of the findings

no comment

Additional comments

no comment

·

Basic reporting

1. The authors used clear and fairly unambiguous professional English language in the study.

2. However, it's of note that the literature review relevant to the Title as is presently embodied in the work is very scanty; this may of course be explained by the fact that the condition, Caesarean Scar Pregnancy (CSP), being researched on is a very rare and uncommon condition.

Experimental design

We wish to observe, with lots of surprise, that the Research Title is incongruent with the Title showcased in the Ethical Approval document. The authors should please reconcile and rectify this major anomaly to avert bias.

Validity of the findings

In lines 60 through 65, the three (3) studies referenced by the authors in this article, which seem to harbor the main thrust and focus of the work they set out to do, are not relevant and not directly related or relevant to their study on CSP.

Generally, each of the references is essentially and summarily focused on the relationship between insulin resistance of pregnancy and other co-morbidities like gestational diabetes, pre-eclampsia, sterile inflammatory conditions, etc, in pregnancy. None of the cited studies has any bearing on CSP.

One is therefore not surprised to observe that their findings do not have a significant positive validity outcome regarding the aim and research questions implied in the study.

Additional comments

This study can be published after addressing the salient issues and questions raised above.

However, it should be noted that the work, as it presently is, may not likely be seen to have significantly contributed to the burden of knowledge. This is so because pregnancy on its own, even without CSP and its management, is generally known to lead to increased TyG index.

So, even when there is an increase in TyG index, one may not completely attribute it to CSP or the surgical management, as to associate it with their PRRF, since pregnancy on its own can be a reason for a rise in TyG index.

---

## Round 0.3 · accepted · Accept

You have addressed all feedback provided by the reviewers or provided suitable rebuttal where necessary.

·

Basic reporting

The authors used clear and unambiguous English language in the work.

Experimental design

Good. But should ensure that the Title in the Ethical Approval page confirms with the Title of the work.

Validity of the findings

No new comment.

Additional comments

I'm not able to see where the authors change the Title of the Ethical Approval page to conform well with the Title of their research work. If they've done that as I earlier pointed out, then, the work can be accepted for publication-provided other corrections have also been addressed.